# Is Chronic Kidney Disease Due to Cadmium Exposure Inevitable and Can It Be Reversed?

**DOI:** 10.3390/biomedicines12040718

**Published:** 2024-03-23

**Authors:** Soisungwan Satarug

**Affiliations:** Kidney Disease Research Collaborative, Translational Research Institute, Woolloongabba, Brisbane, QLD 4102, Australia; sj.satarug@yahoo.com.au

**Keywords:** albuminuria, β_2_-microglobulin, cadmium, chronic kidney disease, GFR, health risk, protein reabsorption

## Abstract

Cadmium (Cd) is a metal with no nutritional value or physiological role. However, it is found in the body of most people because it is a contaminant of nearly all food types and is readily absorbed. The body burden of Cd is determined principally by its intestinal absorption rate as there is no mechanism for its elimination. Most acquired Cd accumulates within the kidney tubular cells, where its levels increase through to the age of 50 years but decline thereafter due to its release into the urine as the injured tubular cells die. This is associated with progressive kidney disease, which is signified by a sustained decline in the estimated glomerular filtration rate (eGFR) and albuminuria. Generally, reductions in eGFR after Cd exposure are irreversible, and are likely to decline further towards kidney failure if exposure persists. There is no evidence that the elimination of current environmental exposure can reverse these effects and no theoretical reason to believe that such a reversal is possible. This review aims to provide an update on urinary and blood Cd levels that were found to be associated with GFR loss and albuminuria in the general populations. A special emphasis is placed on the mechanisms underlying albumin excretion in Cd-exposed persons, and for an accurate measure of the doses–response relationships between Cd exposure and eGFR, its excretion rate must be normalised to creatinine clearance. The difficult challenge of establishing realistic Cd exposure guidelines such that human health is protected, is discussed.

## 1. Introduction

Chronic kidney disease (CKD) is a progressive disease which affects 8–16% of the world’s population [1,2,3]. It contributes significantly to global morbidity and mortality, and it is predicted to become the 5th leading cause of years life lost by 2040 [4]. Aging, being overweight, having diabetes or hypertension and chronic exposure to nephrotoxic chemicals are risk factors for CKD [1,2,3]. 

Diagnostic criteria for CKD include a fall in the estimated glomerular filtration rate (eGFR) below 60 mL/min/1.73 m^2^, termed low eGFR, or there is albuminuria that persists for at least 3 months [1,2]. Albuminuria is designated when the excretion of albumin (E_alb_), measured as the albumin-to-creatinine ratio (ACR), rises to levels above 20 and 30 mg/g creatinine in men and women, respectively [1,2].

Cadmium (Cd) is a metal pollutant and a cumulative nephrotoxin presenting worldwide public health concerns because it is found in most food types [5,6,7]. For most people, diet is the main non-workplace exposure to Cd [5,6,7]. Polluted air, active and passive smoking are additional Cd exposure sources [8,9]. The intestinal absorption of Cd and its transport pathways to kidneys and other target organs follow those for essential metals closely, notably iron, zinc and calcium [1,10]. The roles of zinc and iron transporters and proteins of iron homeostasis as the determinants of blood and urinary Cd levels have been substantiated by genetic linkage studies [11,12,13]. Body iron stores and nutritional status of zinc are inversely associated with the body burden of Cd [14].

Due to a lack of excretory mechanisms, the kidney Cd content increases with age [15,16,17]. The most frequently reported effects of Cd exposure include kidney tubular cell damage and tubular proteinuria, evident from an increased excretion of the low-molecular weight protein β_2_-microglobulin (β_2_M) [18]. An increase in β_2_M excretion has long been used as evidence of tubular dysfunction. A Japanese cohort study reported that an increased level of urinary β_2_M was associated with a 79% increased risk of a large decline in eGFR (10 mL/min/1.73 m^2^) over a five-year observation period [19]. Thus, an increased β_2_M may also result from loss of nephrons for any reason.

This review has three major aims: firstly, to provide an update on the exposure levels to Cd in the general population that were found to be associated with an enhanced risk of CKD. A special emphasis is placed on the mechanisms by which Cd reduces the absorption of albumin, leading to albuminuria. This has not been clearly defined. The second aim is to demonstrate the impact of a conventional method of the normalisation of the excretion rates of Cd and albumin to creatinine excretion (E_cr_) on the dose–response and health risk analyses of Cd. The third aim is, to illustrate that current exposure guidelines do not provide enough of a margin to protect health. To this end, eGFR reduction and albuminuria are proposed to be the adverse health outcomes of chronic Cd exposure according to which protective exposure guidelines should be formulated.

## 2. Environmental Cadmium Exposure and Risk of Chronic Kidney Disease 

This section highlights epidemiological data that link an increased risk of CKD to Cd exposure in non-occupationally exposed populations.

### 2.1. Findings from Systematic Reviews and Meta-Analyses 

Doccioli et al. (2024) conducted a systematic review and meta-analysis to evaluate the strength of an association between CKD and Cd exposure [20]. They reported that Cd exposure was associated with an increased risk of CKD only when assessed by eGFR, and the association was more evident for blood than for urinary Cd or dietary exposure [20].

Previously, Jalili et al. (2021) reported that an association of eGFR and urinary Cd was insignificant, but the risk of proteinuria rose by 35%, when the top category of Cd dose metrics was compared with the bottom Cd exposure category [21]. Byber et al. (2016) concluded that Cd exposure was not associated with a progressive decline in eGFR [22]. 

Nearly all studies used ACR to define albuminuria and urinary Cd normalised to E_cr_ as E_Cd_/E_cr_ to indicate long-term exposure to Cd. However, the E_cr_-adjustment appeared to introduce variance to datasets, which created a high degree of statistical uncertainty [23]. In effect, the associations between E_Cd_/E_cr_ and eGFR and E_alb_/E_cr_ (ACR) are statistically insignificant (Section 2.3). 

### 2.2. Exposure Levels of Concern 

The dose–response studies showing Cd exposure levels associated with a low eGFR, albuminuria and proteinuria can be found in Table 1. 

A study from Thailand (n = 1189) reported 6.2-fold and 10.6-fold increases in the risk of a low eGFR as urinary Cd excretion levels rose from ≤0.37 to 0.38–2.49 and ≥2.5 µg/g creatinine, respectively [25]. In a study from China, there was a 2.98-fold increase in the risk of elevated albumin excretion as urinary Cd levels rose from ≤0.32 to >1.72 µg/g creatinine [29]. In a study from Spain, 1.58-fold and 4.54-fold increases in the risk of albuminuria were associated with urinary Cd levels > 0.27 and >0.54 µg/g creatinine, respectively [30].

An association of CKD with Cd exposure was observed in U.S. population studies, [30,31,32,33]. Specifically, an increased risk of CKD among U.S. citizens was linked to blood Cd levels ≥ 0.6 µg/L and urinary Cd levels ≥ 1 µg/g creatinine. Deaths from all causes among U.S. citizens who had CKD increased linearly with urinary and blood Cd levels [33]. 

The median for E_Cd_/E_cr_ in women enrolled in the NHANES 1988–1994 was 0.77 μg/g creatinine, higher than that of men (0.58 μg/g creatinine) [34]. A study from Taiwan, including 977 men and 1470 women (mean age 55), reported a mean urinary Cd concentration higher in women than men (0.9 vs. 0.7 µg/L) [35]. A study from Sweden observed higher blood Cd in women than men of a similar age [36]. In a population-based study of Chinese subjects, aged 2.8 to 86.8 years (n = 1235), urinary Cd levels increased with age, peaking at 50 and 60 years in non-smoking women and men, respectively [37].

Associations of eGFR and Cd exposure indicators have been observed in both children and adult populations. Studies from Guatemala [38] and Myanmar [39] reported associations of lower eGFR values and higher Cd excretion rates. A prospective cohort study of Bangladeshi preschool children reported inverse relationships of kidney volume and eGFR with Cd excretion in children at age 5 years [40]. Urinary Cd excretion showed an inverse association with eGFR, especially in girls [40]. A prospective cohort study of Mexican children reported that the mean for dietary Cd exposure was 4.4 µg/d at the baseline, and it rose to 8.1 µg/d after nine years, when a marginally inverse association of eGFR and dietary Cd exposure was observed [41].

The data in Table 1 indicate that E_Cd_/E_cr_ values ≥ 0.27–0.32 µg/g creatinine may be sufficient to contribute to increased albumin excretion and a decline in eGFR. These responses appeared to occur long before E_Cd_/E_cr_ reached 5.24 μg/g creatinine, a level at which E_β2M_/E_cr_ was ≥300 μg/g creatinine. Thus, measures of CKD may serve as suitable indicators of adverse health effect and a basis on which protective exposure guidelines should be formulated (Section 4). 

In summary, studies from various countries report disparate levels of urinary and blood Cd. However, they are broadly consistent in reporting that urinary Cd levels associated with a low eGFR and albuminuria were below 5.24 µg/g creatinine, which is the current nephrotoxicity threshold level for Cd. This Cd toxicity threshold level was obtained from a risk assessment model that used β_2_M excretion ≥ 300 μg/g creatinine as an indicator of adverse health effect of exposure to Cd, detailed in Section 5. 

### 2.3. Impact of Normalisation of Urinary Excretion Rates of Cadmium and Albumin 

In the most recent meta-analysis [20], Cd exposure was associated with a low eGFR, but not albuminuria. In this Section, a practice of adjusting urinary excretion of albumin to E_cr_, known as ACR, is discussed as a reason for such a report of non-association between albuminuria and Cd exposure. 

According to the ACR criterion for CKD diagnosis, a higher ACR cutoff value is used to define albuminuria in women to account for male–female differences in muscle mass. This practice simply reflects the fact that universally women have a lower muscle mass, and thus lower E_cr_, compared to men. For the same reason, the normalisation of Cd excretion to E_cr_ yields typically higher E_Cd_/E_cr_ values in women, compared to men of similar age [34,35,36,37]. 

The normalisation of E_Cd_ to the excretion of creatinine (E_cr_) corrects for urine dilution as follows. If V_u_ is the rate of urine flow, E_Cd_ and E_cr_ equal [Cd]_u_V_u_ and [cr]_u_V_u_, respectively, and E_Cd_/E_cr_ simplifies to [Cd]_u_/[cr]_u_. Since these two variables, [cr]_u_ and [Cd]_u_, are not connected biologically, the ratio does not normalise [Cd]_u_ to a factor that affects E_Cd_. The sole virtue of [Cd]_u_/[cr]_u_, as opposed to [Cd]_u_ alone, is that it adjusts [Cd]_u_ for V_u_. However, this adjustment introduces a source of imprecision, because E_cr_ is proportional to muscle mass, which varies among people.

Similarly, the normalisation of E_Cd_ to creatinine clearance, E_Cd_/C_cr_ is determined by calculation using an equation [Cd]_u_[cr]_p_/[cr]_u_, which is algebraically simplified from [Cd]_u_V_u_/[cr]_u_V_u_/[cr]_p_, where p = plasma and u = urine; E_Cd_ = urinary excretion rate of Cd; V_u_ = urine flow rate; cr = creatinine [42]. E_Cd_/C_cr_ is expressed as µg/L of filtrate.

The GFR is the product of the nephron number and mean single nephron GFR, while creatinine clearance (C_cr_) approximates the GFR [43,44,45]. Because C_cr_ varies directly with nephron mass, E_Cd_/C_cr_ depicts the burden of Cd per functioning nephron. Because most or all excreted Cd emanates from injured or dying tubular cells [46]. E_Cd_/C_cr_ quantifies the severity of the injury due to Cd accumulation at the present time, not the risk of injury in the future. 

Timed urine collections are not required. Variation in [cr]_u_ with muscle mass does not affect E_Cd_/C_cr_, because [cr]_u_ and [cr]_p_ are proportionally related at any C_cr_. At a given tubular cell Cd content, the effect of a reduced nephron number to a lower E_Cd_ is offset in the calculation by a rise in [cr]_p_ as C_cr_ falls, and excretion of Cd per intact nephron is accurately depicted. 

### 2.4. Demonstrable Dose–Response Relationships 

To show the effects of normalisation methods on dose–effect relationship evaluation, data on urinary excretion of Cd and various proteins recorded for residents of a Cd-contaminated area of Thailand [47] are recapitulated (Table 2).

Among 215 study subjects, 33, 131 and 51 had eGFR > 90, 61–90 and ≤60 mL/min/1.73 m^2^, respectively (Table 2). There was no variation in the urinary excretion of Cd, as E_Cd_/E_cr_, across the three eGFR groups, thereby suggesting a non-association of eGFR and Cd exposure. In comparison, the urinary excretion of Cd, as E_Cd_/C_cr_, differed across the three eGFR groups; the mean E_Cd_/C_cr_ was the highest, middle and lowest in the low-, moderate- and high-eGFR groups. Those in the low-eGFR group excreted β_2_M, α_1_M, albumin, total protein and Cd at the highest rates. 

Thus, an accurate quantification of an effect of Cd on the glomerular and tubular functions of kidneys can only be realised when the excretion rates of β_2_M, α_1_M, albumin, total protein and Cd itself are normalised to C_cr_. The C_cr_ normalisation is not superfluous given that it eliminates conceptual flaw in the adjustment of the excretion rate to creatinine excretion. 

As nephrons are lost, the amount of Cd excreted is expected to be reduced. However, the effect of a reduced nephron number to a lower E_Cd_ is offset in the C_cr_ normalisation by a rise in plasma creatinine as C_cr_ falls. Consequently, the excretion of Cd per intact nephron is accurately depicted. 

In summary, C_cr_ normalisation is not affected by muscle mass while it corrects for differences in urine dilution and the number functioning nephrons. The utility of C_cr_ normalised data in mechanistic dissection of the effects of Cd are indicated also in Table 3 and Table 4 [49].

No dose–effect relationships were observed between E_Cd_/E_cr_ and the prevalence odds of albuminuria and β_2_-microglobulinuria (Table 3). There was a 3.5-fold increase in the risk of a low eGFR in those with Cd excretion rates of 5–9.99 µg/g creatinine, but not the most severely affected group, compared with a Cd excretion rate < 2 µg/g creatinine. In comparison, clear dose–response relationships were observed for all three adverse outcomes (Table 4). Thus, in Cd-exposed subjects, normalisation of excretion rates to C_cr_ demonstrated dose–effect relationships that were not evident with normalisation to E_cr_. 

Based on C_cr_-normalised data, an increased excretion of albumin and β_2_M and a eGFR decline appeared to occur simultaneously. Among these outcomes, eGFR was affected the most, while albumin and β_2_M were similarly affected. Notably, the eGFR effect of Cd was obscured completely when E_Cd_ was normalised to E_cr_ (Table 2 and Table 3). For C_cr_-normalised data, the respective prevalence odds ratios for a low eGFR rose 5.7-fold, 10.3-fold and 18.1-fold in those with (E_Cd_/C_cr_) × 100 values of 2–4.99, 5–9.99 and ≥10 µg/L of filtrate, respectively, compared with (E_Cd_/C_cr_) × 100 values below 2 µg/L of filtrate.

Clearly, a practice of adjusting the urinary excretion rate of albumin and Cd to E_cr_ underestimated the severity of Cd effects on both glomerular and tubular functions. 

## 3. Cadmium and Albuminuria 

This section provides a summary of the current understanding of tubular reabsorption of proteins. The mechanisms by which Cd reduces albumin reabsorption are highlighted together with the functional consequences of renal Cd accumulation, quantified by fractional reductions in the reabsorptions of albumin. The implication of albumin reabsorption for the delivery of Cd to proximal tubules is discussed.

### 3.1. Tubular Protein Reabsorption: Overview 

Blood perfuses the kidneys at the rate of 1 L per minute, and all renal blood flow is directed through afferent arterioles into glomeruli [50]. Under normal physiologic conditions, 20% of plasma entering the glomerulus is filtered into Bowman’s space. At least 90% of the circulating protein is ultrafilterable, and 99.9% of the filtered load is reabsorbed [51,52,53]. Approximately 40–50 g of protein can be retrieved each day in the proximal tubule of the kidneys, which is divided into segments S1, S2 and S3 [54,55,56,57,58]. The reabsorption of protein via receptor-mediated endocytosis (RME) involving megalin and cubilin occurs mostly in S1, whereas fluid phase endocytosis (FPE) occurs in all three segments [59,60].

Impaired tubular reabsorption of proteins is a known sign of Cd intoxication, which is reflected by an increased excretion of the low-molecular-weight proteins, namely retinol binding protein, α_1_M, and β_2_M, reviewed in Satarug and Phelps, 2021 [61]. Given that Cd intoxication is known to impair the reabsorption of β_2_M (Table 2), it was hypothesised that Cd may interfere with the reabsorption of albumin as well (Figure 1).

In normal kidney health, filtered β_2_M is reabsorbed and degraded mostly in S1 and to a lesser extent in S2 [62]. β_2_M is a constituent of the neonatal Fc receptor, FcRn, which mediates the transcytosis of reabsorbed albumin [18,63,64]. There is little evidence for the transcytosis of β_2_M. Albumin reabsorption occurs in S1, S2 and S3 [58,65], and FPE is believed to initiate most of the transcytosis of albumin [54]. 

In the circulation, most Cd (90%) is bound to haemoglobin in red blood cells [66,67]. The remainder of Cd (10%) is found in plasma, associated with albumin, histidine and other non-protein thiols, including glutathione, cysteinylglycine, homocysteine and γ-glutamylcysteine [67,68]. The total plasma concentrations of albumin thiols and non-protein thiols were 0.6 mM and 12–20 µM, respectively [68]. As a principal carrier of plasma Cd, the reabsorption of albumin complexed with Cd may provide Cd an entry route to PTCs. In an in vitro experiment, cell injury was observed in the rat proximal tubule WKPT-0293 Cl.2 cells exposed to albumin and β_2_M complexed with Cd, but the injury was not evident when cells were exposed to albumin or β_2_M alone [69]. 

Because the binding of Cd alters the conformation of albumin [70], Cd-bound albumin probably undergoes RME by the megalin–cubilin system and subsequent lysosomal degradation. Cd released during this process may then disrupt megalin homeostasis.

### 3.2. Albuminuria in Cadmium-Exposed Subjects 

Albumin is a negatively charged protein with a molecular weight of 66 kDa and an average half-life in the plasma of 19 days [56,57,71]. Approximately 10–15 g of albumin is synthesised daily in the liver and released into the circulation [54,57,71]. The plasma concentration of albumin ranged between 3.5 g/dL and 5 g/dL. Normally, albumin is not filtered by glomeruli due to its negative charge and a high molecular weight [71]. However, 1–10 g of albumin may reach the tubular lumen by means of transcytosis through endothelial cells and podocyte foot processes [72,73]. The degradation of albumin occurs mainly in muscle, liver and kidney proximal tubular epithelial cells [71]. 

Albuminuria was observed in female Wistar rats exposed to Cd at 3 mg/kg/day by gavage for 8 weeks [74]. It was suggested that such albuminuria was due to a decreased level of cubilin protein which impaired the tubular reabsorption of albumin [74]. Studies using human renal glomerular endothelial cells showed that Cd at a non-cytotoxic level (1 µM) may increase the glomerular permeability to albumin by causing the redistribution of adheren junction protein vascular endothelial cadherin and β-catenin [75,76]. Another study using cultured LLC-PK1 cells and pig proximal tubular cells showed that Cd reduced the levels of megalin and chloride channel 5 (ClC5), thereby decreasing albumin reabsorption [77]. 

In any mechanistic dissection, a clear dose–response relationship must first be established, and a population exposed to a wide range of Cd doses is required to meet this requirement. The Mae Sot District in western Thailand appeared to be ideal because it was an area where environmental Cd pollution was endemic [78,79]. This geographic area provided a well-circumscribed population of people with the same level of exposure that would enable one to discern the health impact of dietary Cd exposure [80,81]. More than 40% of residents aged ≥40 years were at risk of Cd-induced toxic injury and Cd-induced tubular dysfunction [81]. Furthermore, the level of Cd exposure among the Mae Sot residents appeared to be moderate enough to be likely experienced by many populations.

In a logistic regression analysis of data from the Mae Sot residents (Table 4), the risk of albuminuria rose 2.1–7.9% for every one-year increase in age; it also rose 2-fold, and 1.9-fold in smokers and those with hypertension. In comparison, the risk of β_2_-microglobulinuria was not affected by age, smoking or hypertension. However, β_2_-microglobulinnura and albuminuria both were related to the excretion of Cd in a dose-dependent manner. Adjustment for age reduced variance and tightened the correlations of albumin excretion, β_2_M excretion and Cd excretion. Consequently, the slope of albumin excretion vs. β_2_M excretion regressions approached unity. Apparently, these data indicated that Cd affected a single mechanism, leading to reduction in the reabsorption of both albumin and β_2_M. The proposed mechanism of Cd-induced albuminuria is depicted in Figure 2.

In S1, β_2_M is reabsorbed via RME, involving the apical protein megalin. β_2_M is then degraded in lysosomes. Albumin is reabsorbed via RME in S1, and FPE in S1, S2 and S3, and is mostly returned to the circulation via transcytosis. A small fraction of albumin, reabsorbed through RME, undergoes lysosomal degradation. Cd does not disrupt the transcytosis of albumin, but it impairs a single mechanism for RME and the degradation of both β_2_M and albumin. As inferred from the literature reports, it is proposed that Cd disrupts particularly the function of megalin, thereby decreasing the reabsorption rates of both proteins [49]. 

### 3.3. Fractional Reductions in the Reabsorption of Albumin and β_2_M 

Fractional reductions in the reabsorption of albumin and β_2_M were estimated to assess the functional consequences of renal Cd accumulation for protein reabsorption [49]. 

In the least affected subjects (eGFR > 90 mL/min/1.73 m^2^), the mean E_alb_/C_cr_ was 8.57 × 10^−2^ mg/L of filtrate and the mean E_β2M_/C_cr_ was 5.97 µg/L of filtrate. In the most affected subjects (eGFR < 60 mL/min/1.73 m^2^), the corresponding values of E_alb_/C_cr_ and E_β2M_/C_cr_ were 70.27 × 10^−2^ mg/L of filtrate and 411 µg/L of filtrate, respectively. In the latter group, the mean eGFR was 46.6 mL/min or 67.1 L/d/1.73 m^2^. 

The fractional excretion of albumin (FE_alb_; excretion rate/filtration rate of albumin) can be estimated as (E_alb_/C_cr_)(eGFR)/(GSC_alb_)([alb]_p_])(eGFR), or (0.7027 mg/L of filtrate)(67.1 L/d)/(10^−2^)(40,000 mg/L)(67.1 L/d) = 0.0018, or 0.18%, if a glomerular sieving coefficient for albumin (GSC_alb_) of 10^−2^ and plasma albumin concentration ([alb]_p_) of 40 gm/L are assumed. This means that the mean fractional tubular reabsorption of albumin (FTR_alb_) was 99.8%, even though a rise in absolute albumin excretion was discernible as the eGFR fell. If GSCalb is assumed to have been 10^−4^ instead of 10^−2^, the FE_alb_ is 18% and FTR_alb_ is 82%. 

If a GSC_β2M_ of 1 and [β_2_M]_p_ of 2.0 mg/L (2000 µg/L) are assumed, the mean F_Eβ2M_ is (E_β2M_/C_cr_)(eGFR)/(GSC_β2M_)([β_2_M]_p_)(eGFR), or (411 µg/L of filtrate)(67.1 L/d)/(1)(2000 µg/L of plasma)(67.1 L/d) = 0.2055, or 21%. This means that the FTR_β2M_ is 79%. 

It is noteworthy that although the reductions are likely to have resulted from the same altered mechanism, fractional reductions in the reabsorption of albumin and β_2_M differ greatly if the GSC_alb_ of 10^−2^ is assumed, and is similar if a GSC_alb_ of 10^−4^ is assumed.

### 3.4. Overall Effects of Cadmium Burden on Tubular Function 

To assess the overall impact of Cd on tubular protein reabsorption function, the amounts of albumin and β_2_M that were reabsorbed through RME and subjected to catabolism in lysosomes were estimated, assuming glomerular sieving coefficients for albumin (GSC_alb_) and β_2_M (GSC_β2M_) to be 0.01 and 1, respectively (Table 5).

### 3.5. Implication of Albumin Reabsorption for the Delivery of Cadmium to Proximal Tubules

The evolving concept that proximal tubules reabsorb dozens of grams of albumin per day (Table 5) raises important theoretical possibilities concerning the access of Cd to tubular cells. Red blood cells (RBCs) carry at least 90% of circulating Cd [66]. In a lysate of rabbit RBCs, the metal associated primarily with haemoglobin and glutathione [67] in the lysates of RBCs from mice pretreated with subcutaneous Cd for six months, the metal was bound to haemoglobin and to a smaller species, probably MT [66]. Multiple investigative techniques have shown that Cd binds to specific sites on the globin chains of haemoglobin, represented in Figure 1 [82].

The normal mean lifespan of RBCs (and therefore haemoglobin) is 120 days, but changes in RBC membranes induced by Cd may alter the shape of cells, induce premature haemolysis in the reticuloendothelial system (RES), and thereby shorten cellular lifespan [83,84,85]. When senescent RBCs are destroyed in the RES, heme porphyrin groups are metabolised to bilirubin, which is taken up by circulating albumin [86,87]. Presumably, Cd is simultaneously released from globin chains as they are broken down to their constituent amino acids, and since albumin is so abundant in plasma, it is speculated that it is the principal scavenger of Cd from the RES. Cd–albumin complexes are continuously presented to hepatocytes and proximal tubular cells at high rates of blood flow, and both cell types store Cd acquired as CdMT [61,88,89]. The internalisation of Cd from albumin complexes by hepatocytes has been shown [90].

Because the binding of Cd alters the conformation of albumin [70,91], filtered Cd–albumin complexes probably undergo RME by the megalin–cubilin system and subsequent lysosomal degradation. Cd released during this process may then disrupt megalin homeostasis. If the above proposal is correct, then most Cd assimilated from exogenous sources is destined to interact with albumin eventually, even if it is bound to haemoglobin initially.

### 3.6. Summary on the Impact of Cadmium on Protein Reabsorptive Function 

The accumulation of Cd in the kidneys reduced receptor-mediated endocytosis of albumin and β_2_M. Estimated fractional reductions in the reabsorptions of albumin and β_2_M were similar (18 vs. 21%) assuming the glomerular sieving coefficients for albumin and β_2_M to be 10^−4^ and 0.01, respectively. These impacts of Cd were quantifiable because of the clear dose–effect relationships of E_alb_/C_cr_, E_β2M_/C_cr_, eGFR and the nephron burden of Cd, indicated by E_Cd_/C_cr_. In contrast, E_alb_/E_cr_ (ACR), E_β2M_/E_cr_ and E_Cd_/E_cr_ were unrelated, thereby precluding a dose–response analysis and nullifying the quantification of Cd effects.

## 4. eGFR Decline and the Health Risk Assessment of Environmental Cadmium 

The data in Table 1 indicate that urinary Cd levels associated with a low eGFR were below the current nephrotoxicity threshold level for Cd of 5.24 µg/g creatinine. Arguably, a declining eGFR may serve as a suitable indicator of the adverse health effect of Cd exposure and a basis to derive protective exposure guidelines. In this section, data on the kidney burden of Cd and an inverse relationship between eGFR and urinary excretion rate of Cd are reiterated together with a benchmark dose (BMD) estimate of the Cd burden that may carry a discernible adverse effect. This BMD estimate has now become a replacement of no-observed-adverse effect level (NOAEL) [92,93]. 

NOAEL is referred to the highest experimental dose level for which the response does not significantly differ from the response in the control group [94]. 

### 4.1. Measurement of Kidney Burden of Cadmium 

Cd accumulation in human kidneys can be found in reports of the analysis of post-mortem and biopsied samples (Table 6).

Of note, the Australian study measured the lung Cd content, which was used to assess the contribution of inhalational exposure, where females were found to have higher hepatic and renal cortical Cd levels than males after adjustment for age and inhalational exposure. Renal cortical Cd content increases progressively until the age of 50 years, and declines sharply thereafter. The peak kidney Cd content was 25.9 µg/g [15]. 

The hepatic Cd content increased gradually with age without interruption, and it was higher in women than in men. It is speculated that iron depletion due to menstrual losses promoted intestinal Cd absorption in women during premenopausal years. It is also speculated that nephron loss and interstitial scarring due to aging and Cd toxicity caused the observed decline in cortical Cd content after the age of 50 years [100,101].

### 4.2. Cadmium Excretion and Glomerular Filtration Rate

A paradox is evident in the reported relationships of GFR and environmental Cd exposure. Some investigators found that E_Cd_ rose with the GFR when exposure was low [102,103,104]. At least two groups found that the GFR fell from normal values as E_Cd_ rose minimally [105,106]. To reconcile these observations, it was speculated that the nephrotoxicity of Cd begins with a transitory phase in which cell injury is releasing Cd to filtrate but has not yet led to cell death; during that phase, the number of nephrons determines the E_Cd_. As Cd begins to destroy tubular cells, E_Cd_ increases further, even though nephrons drop out and the GFR begins to decline. 

A large body of work shows that the GFR fell in both occupational and environmental exposure settings. Nephron loss was most extreme in polluted regions of Japan [107], but it was also documented in other Asian countries and Europe, listed in Table 1 [108,109]. The progression of CKD often continued after cessation of exogenous exposure [110].

Reductions in GFR due to Cd nephropathy could be attributed mostly to tubular injury that disables glomerular filtration and ultimately leads to nephron atrophy, glomerulosclerosis, and interstitial inflammation and fibrosis [111,112]. 

Under chronic exposure conditions, cellular defence mechanisms are often overwhelmed, i.e., the binding sites are saturated, and Cd that eludes MT complexation promotes the synthesis of reactive oxygen species (ROS) that inflict injury. This injury induces the apoptosis and necrosis of tubular cells, and it undermines the adhesion of cells to one another. Cellular injury also leads to the release of proteins and CdMT into filtrate; compromises protein reabsorption (Section 2) and substances co-transported with sodium; and ultimately reduces the GFR through the destruction of nephrons.

### 4.3. The NOAEL Equivalent Value of Cadmium Burden 

Using eGFR decline as an adverse health effect, the NOAEL equivalent value of E_Cd_/C_cr_ was determined from a cohort of 1189 Thai subjects (493 males and 696 females) with a mean age of 43.2 years [24]. By definition, a NOAEL equivalent is the lower 95% confidence bound of the benchmark dose (BMD), termed the BMDL value, which is derived from fitting data to various dose–response models, and the E_Cd_/C_cr_ value that produces eGFR reduction less than 5% can be considered the level that carries a negligible health risk [93,94,113].

The overall percentages of smokers, subjects with hypertension and low eGFR in the Thai cohort used to determine the NOAEL equivalent values of E_Cd_/C_cr_ were 33.6%, 29.4% and 6.2%, respectively. The dose–response models and the lower and upper 95% confidence bounds of the BMD (BMDL/BMDU) computed from 5% eGFR reduction are provided in Figure 3.

The NOAEL equivalent value of (E_Cd_/C_cr_) × 100 was 1.13 µg/L of filtrate in both men and women. Because the basic cellular toxicity mechanism of Cd should be the same, the NOAEL for any effect of Cd can be expected to be identical in women and men. This estimate of a NOAEL equivalent can be translatable to E_Cd_/E_cr_ values between 0.01 and 0.02 µg/g creatinine.

A previous risk analysis, using β_2_-microglobulinuria as an indicator of adverse health effect, reported that a BMD value of Cd excretion was higher in female than male residents (8.1 vs. 6.9 μg/g creatinine [81]). These figures are more than 500 times higher than the NOAEL equivalent computed from eGFR decline endpoint. 

### 4.4. Evidence for the Threshold Level of Cadmium 

A recent dose–response analysis revealed that a non-toxic level of Cd burden was extremely low (Figure 4), in ranges with the NOAEL equivalent obtained above. 

Scatterplots relating eGFR to E_Cd_/C_cr_ showed that the eGFR rose with Cd exposure when (E_Cd_/C_cr_) × 100 values were ≤1.44 and 1.25 µg/L filtrate in women and men, respectively (Figure 4a,b). An inverse relationship of eGFR and Cd exposure was evident when (E_Cd_/C_cr_) × 100 values rose above 1.44 and 1.25 µg/L filtrate in women and men, respectively (Figure 4c,d). 

## 5. Past and Present Health Threat of Environmental Cadmium 

An outbreak of severe Cd poisoning, known as “itai-itai” disease [115], has brought into focus health threats from the consumption of rice heavily contaminated with Cd. To safeguard against excessive dietary exposure, a tolerable intake level of Cd, a reference dose (RfD), toxicological reference value and permissible levels of Cd in foods were determined [116,117]. 

This section gives a summary of the existing dietary Cd exposure guidelines and the nephrotoxicity threshold level, which are not protective of human health. The molecular basis for the cytotoxicity of Cd, especially in the proximal tubular cells, is highlighted.

### 5.1. The WHO Exposure Guidelines and the Nephrotoxicity Threshold Level

Using the excretion rate of β_2_M ≥ 300 μg/g creatinine as an indicator of the nephrotoxicity of Cd, a provisional tolerable weekly intake (PTWI) of Cd at 7 µg per kg body weight per week was derived by the Food and Agriculture Organization and World Health Organization (FAO/WHO) Joint Expert Committee on Food Additives and Contaminants (JECFA) [116]. Later, the PTWI was amended to a tolerable monthly intake (TMI) of Cd at 25 μg per kg body weight per month, equivalent to 0.83 μg per kg body weight per day (58 µg/day for a 70 kg person), and the Cd excretion of 5.24 μg/g creatinine was employed as a nephrotoxicity threshold value [116]. 

The rice Cd content of 0.27 mg/kg may be sufficient to cause kidney and bone damage like those found in itai-itai disease patients [118]. This rice Cd content is below the Codex standard for rice of 0.4 mg/kg [117]. Also, a lifetime Cd intake ≥ 1 g, which is half of the JECFA exposure guideline [116], yielded a 49% increase in mortality from kidney failure, especially among women [119]. These findings cast considerable doubt on the Codex maximally permissible Cd level in rice of 0.4 mg/kg and the lifetime tolerable Cd intake of 2 g [116]. 

A study from China suggested an upper limit of a permissible level of Cd in rice to be 0.2 mg/kg, one half of the Codex standard [120]. All these estimations relied on the threshold of nephrotoxicity at a Cd excretion rate of 5.24 μg/g creatinine to indicate a toxic Cd accumulation level. Of note, the prevalence odds for a low eGFR and albuminuria rose at the Cd excretion levels of ≥0.27–0.32 µg/g creatinine (Table 1). Thus, these indicators of Cd toxicity occurred long before Cd excretion reached 5.24 μg/g creatinine level, at which the excretion of β_2_M rises above 300 μg/g creatinine. Furthermore, the likelihood of eGFR to fall below 60 mL/min/1.73 m^2^ rose 4.7-fold, 6.2-fold and 10.5-fold in those who had β_2_M excretion levels of 100–299, 300–999 and ≥1000 μg/g creatinine, respectively [121]. Because eGFR values ≤ 60 mL/min/1.73 m^2^ are indicative of substantial nephron loss [121], a rise of β_2_M excretion levels ≥ 300 μg/g creatinine is a manifestation of severe toxicity of Cd; as such, its use in health risk estimation is inappropriate. The reasons for a massive increase in β_2_M excretion, especially in those with a low eGFR, can be found below.

### 5.2. β_2_-Microglobulinuria Is the Manifestation of Severe Kidney Pathologies 

β_2_M is a low-molecular-weight protein which is completely filtered by the glomeruli, and virtually all filtered β_2_M is reabsorbed by proximal tubules and subjected to lysosomal degradation [18,122,123,124]. Most of the β_2_M that undergoes tubular degradation is from glomerular filtration, while a small fraction of the protein is from peritubular capillaries [123]. When the GFR drops, the plasma β_2_M level increases subsequently, and an equilibrium between the filtered β_2_M and its tubular degradation is maintained [122,123,124].

Using the same concept as that in Section 3.3, a substantial increase in the excretion of β_2_M in response to nephron loss is demonstrable in the following two scenarios. 

At a β_2_M excretion rate of 300 µg/day, an excretion of Cd of 1 µg/day, GFR of 144 L/day (100 mL/min) and filterable plasma concentration of β_2_M of 2.0 mg/L, the fractional excretion of β_2_M is 0.1% and fractional reabsorption is 99.9%. 

A two-fold increment of urinary excretion of β_2_M to 600 µg/g creatinine necessitates an increase in the fractional excretion of β_2_M from 0.1% to 0.2% and a reduction in fractional excretion of β_2_M to 99.8%. Thus, miniscule Cd-induced reductions in the fractional reabsorption of β_2_M result in an increase in the excretion of β_2_M from 300 to 600 µg/g creatinine [125]. The slight reductions in the fractional reabsorption of β_2_M should not be interpreted to suggest that the effect of Cd is trivial. The excretion rates of β_2_M above 300 µg/g creatinine are at least 10 times higher than in normal populations [126,127].

The excretion of β_2_M can be related to its tubular maximal reabsorption (Tmβ_2_M). This Tmβ_2_M was not found in dogs [123] but in rats, where Tmβ_2_MG was observed to be at a plasma β_2_M concentration four times higher than the normal level [51].

Multiple lines of evidence suggest that Cd may impose a Tmβ_2_M or reduce one that already exists, and an increased excretion of β_2_M indicates reduced β_2_M reabsorption per nephron at any GFR [128]. Once Cd has established a Tmβ_2_M, the excretion of β_2_M can be expected to rise substantially as the GFR falls [122].

### 5.3. Cadmium and Renal Tubular Cell Death: Molecular Basis

#### 5.3.1. Mitochondrial Dysfunction and Calcium Homeostasis Disruption 

Proximal tubular epithelial cells of the kidneys are highly susceptible to mitochondrial toxicant, like Cd, because of their intense metabolic activity, high-energy demand and heavy reliance on autophagy for survival [129,130]. Examples of the effects of Cd on mitochondrial function include impaired energy production, disrupted electron transport chain and enhanced ROS generation [5,130].

The disruption of calcium homeostasis has been suggested to be another mechanism by which Cd induces tubular cell death [131,132,133]. In primary rat proximal tubular cells, Cd caused calcium release from the endoplasmic reticulum, which raised intracellular calcium concentrations and inhibited autophagy [131,132]. In mouse renal tubular cells, Cd appeared to activate at least two calcium channels, namely phospholipase C (PLC)-inositol 1, 4, 5-trisphosphate receptor (IP3R) and the sarco/endoplasmic reticulum Ca2^+^-ATPase (SERCA), known to be involved in the release and reuptake of calcium by the ER [132]. Cd also suppressed SERCA expression and diminished the stability of the SERCA protein in mouse tubular epithelial cells, thereby intensifying the toxic effects of Cd [133].

#### 5.3.2. The Deprivation of Cellular Antioxidant Defence by Cadmium 

At least two major lines of cellular defence mechanisms against the cytotoxicity of Cd have been investigated. One involves the induction of the metal binding protein MT, and the other involves glutathione together with various antioxidant enzyme systems. The induction of heme oxygenase-1 (HO-1) has long been the therapeutic target of Cd toxicity in the belief that the induced expression of HO-1 will generate a potent antioxidant molecule, bilirubin, to neutralise ROS. Cd, itself, is a well-known HO-1 inducer, but distinctively, HO-1 induction by Cd is not coupled with bilirubin synthesis [134].

Takeda et al. (2015) used the eel fluorescent protein UnaG, capable of binding unconjugated bilirubin, to measure intracellular bilirubin production [135]. They found that all cell types synthesised heme as a precursor substrate for the production of antioxidant bilirubin [134]. Takeda et al. have further unveiled that Cd^2+^ and inorganic arsenic as As^3+^ increased the expression of HO-1, but the production of bilirubin remained unchanged. 

In summary, the discovery that HO-1 induction by Cd is not coupled with bilirubin synthesis is of significance to our understanding of the pervasiveness of Cd toxicity. This metal Cd increases cellular oxidative stress and deprives the cellular frontline antioxidant defence at the same time. In the best interest of public health protection, an acceptable Cd exposure level and its toxicity threshold level should be formulated using the current scientific knowledge, elaborated herein. 

## 6. Conclusions

As the result of cadmium accumulation in kidney proximal tubular cells, the GFR, and fractional reabsorption rates of albumin and β_2_-microglobulin are reduced simultaneously, leading to an increased excretion of both proteins. It appears that cadmium adversely affects a single phenomenon involved in the reabsorption of both albumin and β_2_-mcroglobulin. The affected phenomenon is probably receptor-mediated endocytosis involving megalin. 

The practice of adjusting the excretion rates of cadmium and albumin to the excretion of creatinine incorporates a conceptual flaw that can be eliminated if the rates are normalised to creatinine clearance. The urinary excretion rate of cadmium, normalized to creatinine clearance (E_Cd_/C_cr_), quantifies the severity of the kidney tubular injury due to cadmium accumulation at the present time, not the risk of injury in the future.

The NOAEL equivalent of Cd accumulation levels corresponding to a discernible GFR decline is extremely low. Now is the time to acknowledge that there is no safe level of Cd exposure. At present, no effective chelation therapy exists for the removal of Cd from the kidneys. Commonsense therapeutic measures include the cessation of environmental exposure.

## Figures and Tables

**Figure 1 biomedicines-12-00718-f001:**
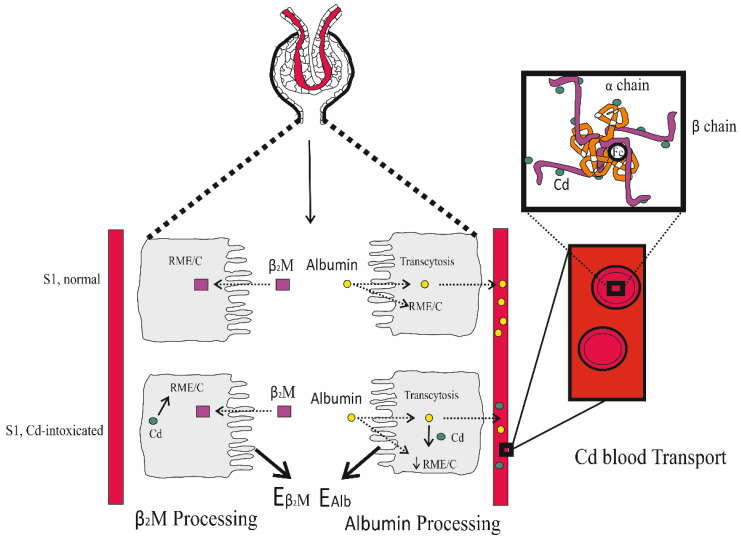
Tubular reabsorption of β_2_-microglobulin (β_2_M) and albumin. β_2_M is reabsorbed by receptor-mediated endocytosis (RME) and is catabolised (C) in lysosomes. Only a small fraction of albumin is reabsorbed via RME. Most albumin is returned to the circulation via transcytosis. Cd intoxication increases the excretion of both β_2_M and albumin. As a carrier of Cd, the reabsorption of albumin may provide a delivery route for Cd to PTCs.

**Figure 2 biomedicines-12-00718-f002:**
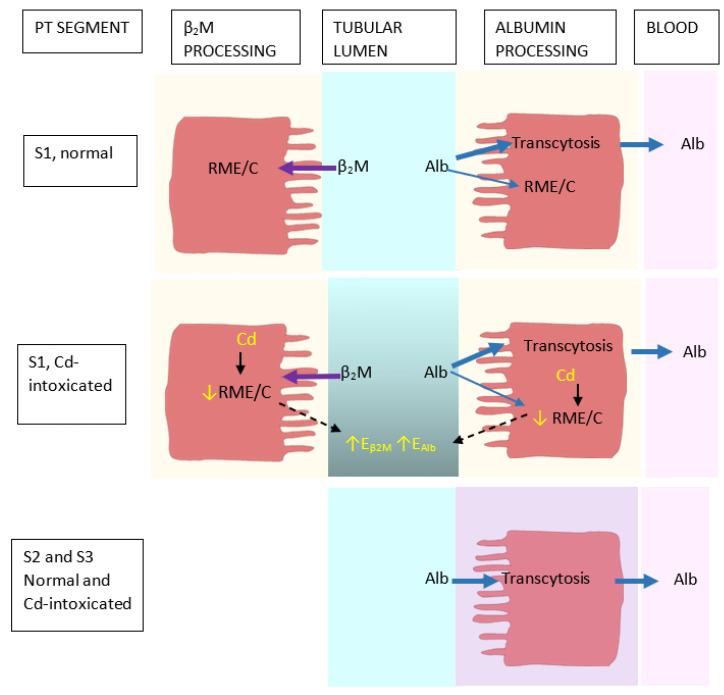
Proposed pathogenesis of reduced proximal tubular reabsorption of albumin and β2-microglobulin after exposure to cadmium. In S1, all β_2_M is reabsorbed through receptor-mediated endocytosis (RME) and is catabolised (C) in lysosomes. Albumin is reabsorbed through RME in S1 and transcytosis in S1, S2 and S3. A small fraction of reabsorbed albumin is subjected to lysosomal catabolism, while most is returned to the blood stream via transcytosis. Cd impairs RME function, which reduces the reabsorption of both albumin and β_2_M.

**Figure 3 biomedicines-12-00718-f003:**
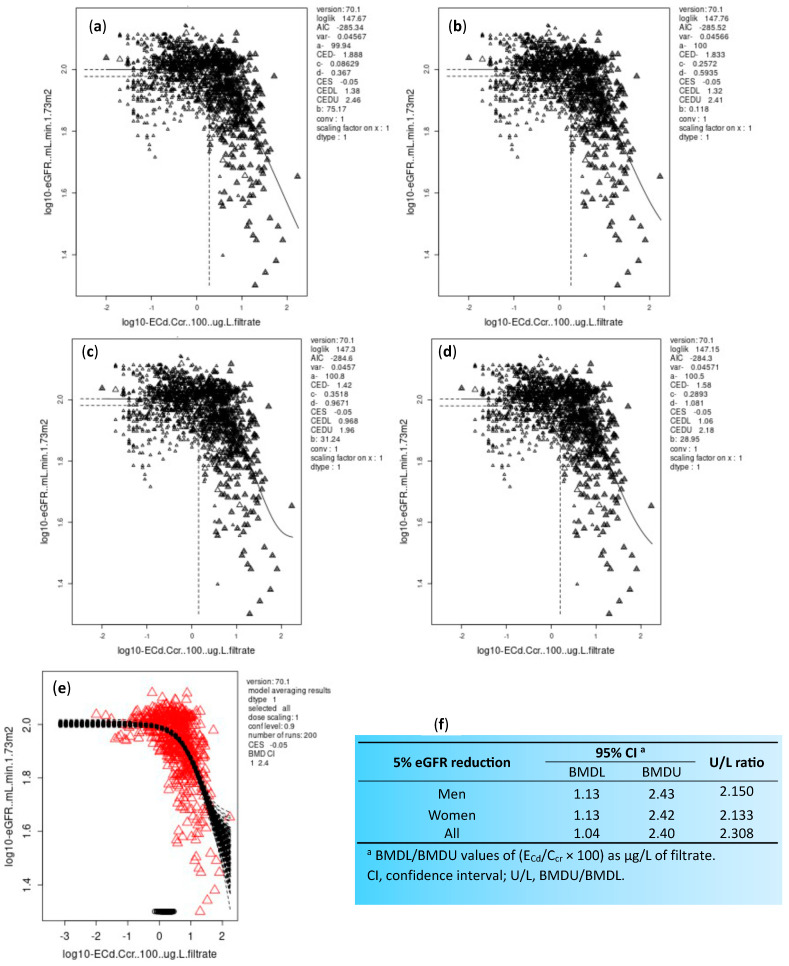
Determination of the NOAEL equivalent of kidney cadmium burden. Pairs of E_Cd_/C_cr_ and eGFR data were fitted to (**a**) an inverse exponential model; (**b**) a natural logarithmic model; (**c**) an exponential model; and (**d**) Hill model. Bootstrap curves for model averaging (**e**). BMDL/BMDU values of E_Cd_/C_cr_ producing a 5% reduction in eGFR (**f**). Data are from Satarug et al., 2022 [24].

**Figure 4 biomedicines-12-00718-f004:**
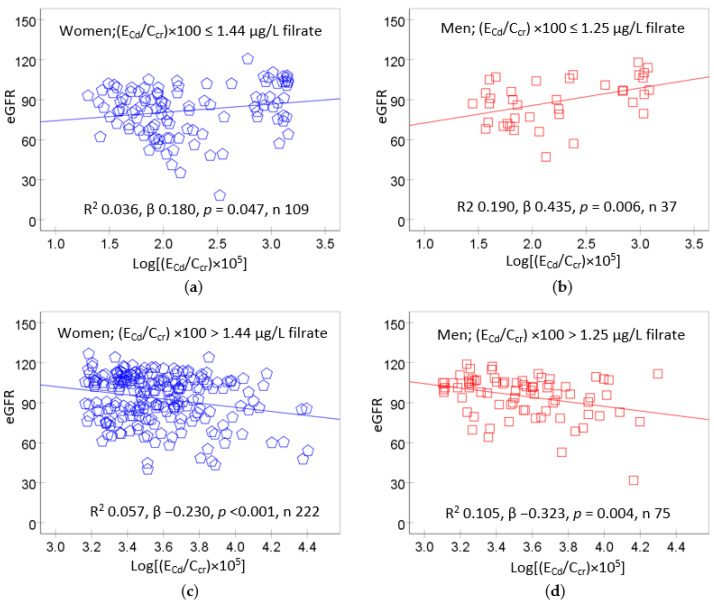
The paradox relationships between the eGFR and urinary excretion rate of cadmium. In women and men with low Cd burdens (**a**,**b**), eGFR showed a positive association with Cd exposure. Conversely, eGFR showed an inverse association with Cd exposure in women and men (**c**,**d**) who had a high Cd burden. Data are from Satarug et al., 2023 [114].

**Table 1 biomedicines-12-00718-t001:** Associations of enhanced risk of CKD with blood and urinary cadmium.

Study Location	Cadmium Exposure Metrics and Effects Observed	Reference
Thailand, n 1189 16–87 years mean age 43.2 years	Risk of low eGFR increased 6.2-fold and 10.6-fold, comparing urinary Cd levels 0.38–2.49 and ≥2.5 µg/g creatinine with ≤0.37 µg/g creatinine, respectively.	Satarug et al., 2022 [24].
Korea, n 2992 20–65 years	Increased risk of low eGFR (OR 1.97) in women was associated with blood Cd levels > 1.74 μg/L.	Myong et al., 2012 [25]
Korea, n 2005 ≥20 years	Increased risk of low eGFR (OR 1.93) was associated with blood Cd in the top quartile (mean, 2.08 μg/L).	Chung et al., 2014 [26]
Taiwan, n 2447 mean age 55.1 years	Increased risk of proteinuria was associated with urinary Cd (OR 2.67) and copper (OR 1.94). Mean urinary Cd in subjects with proteinuria (1.1 μg/L) was 27.3% higher than those without proteinuria.	Tsai et al., 2021 [27]
China n 683 (64.7% women) mean age 57.4 years	Risk of elevated albumin excretion increased 2.98-fold comparing urinary Cd levels ≤ 0.32 with >1.72 µg/g creatinine.	Feng et al., 2022 [28]
Spain, n 1397 age 18–85 years	Increased risks of albuminuria 1.58-fold and 4.54-fold were associated with urinary Cd levels > 0.27 and >0.54 µg/g creatinine, respectively.	Grau-Perez et al., 2017 [29]
United States NHANES 1999–2006 n 14,778, ≥20 years	Blood Cd levels ≥ 0.6 μg/L were associated with a low eGFR (OR 1.32), albuminuria (OR 1.92) and a low eGFR plus albuminuria (OR 2.91).	Navas-Acien et al., 2009 [30]
United States NHANES 1999–2006 n 5426, ≥20 years	Blood Cd levels > 1 µg/L plus urinary Cd levels > 1 µg/g creatinine was associated with albuminuria (OR 1.63). Blood Cd levels > 1 µg/L were associated with a low eGFR (OR 1.48) and albuminuria (OR 1.41).	Ferraro et al., 2010 [31]
United States NHANES 2007–2012 n 12,577, ≥20 years	Blood Cd levels > 0.61 μg/L were associated with a low eGFR (OR 1.80) and albuminuria (OR 1.60). eGFR reduction due to Cd was more pronounced in diabetic and hypertensive people, or both.	Madrigal et al., 2019 [32]
United States NHANES, 1999–2014 A cohort, n 1825 with CKD Follow-up period, 6.8 years	Urinary Cd levels ≥ 0.60 μg/g creatinine were associated with all-cause mortality (HR 1.75; 95%CI: 1.28, 2.39). Blood Cd levels ≥ 0.70 μg/L were associated all-cause mortality (HR 1.59; 95%CI: 1.17, 2.15). Linear dose–response relationships were observed between urinary and blood Cd levels and all-cause mortality.	Zhang et al., 2023 [33]

OR, odds ratio; HR, hazard ratio; CI, confidence interval; NHANES, National Health and Nutrition Examination Survey.

**Table 2 biomedicines-12-00718-t002:** Comparing the excretion rates of various proteins and cadmium in residents of an area of Thailand with endemic cadmium contamination.

Parameter	All Subjects n = 215	eGFR, mL/min/1.73 m^2^			*p*
		>90, n = 33	61–90, n = 131	≤60, n = 51	
**Age, years**	57.0 ± 11.1	49.4 ± 9.4	55.6 ± 9.6	65.6 ± 10.6	<0.001
**BMI, kg/m^2^**	21.4 ± 3.6	21.2 ± 3.2	21.3 ± 3.5	21.7 ± 4.3	0.822
**eGFR, mL/min/1.73 m^2^**	71.6 ± 19.4	100.4 ± 8.3	74.6 ± 8.2	45.4 ± 11.3	<0.001
**Plasma creatinine, mg/dL**	1.07 ± 0.35	0.79 ± 0.13	0.98 ± 0.14	1.50 ± 0.44	<0.001
**Urine creatinine, mg/dL**	118.4 ± 62.2	99.1 ± 53.1	116.8 ± 60.2	135.2 ± 69.4	0.054
**Urine Cd, µg/L**	11.85 ± 12.28	11.18 ± 18.70	10.56 ± 8.05	15.61 ± 15.31	0.079
**Urine β_2_M, mg/L**	4.92 ± 17.43	0.20 ± 0.36	1.18 ± 4.02	17.57 ± 32.31	<0.001
**Urine α_1_M, mg/L**	13.09 ± 18.68	5.66 ± 6.17	8.37 ± 7.91	30.04 ± 30.31	<0.001
**Urine albumin, mg/L**	25.57 ± 70.59	7.62 ± 7.29	22.64 ± 76.57	44.72 ± 73.74	<0.001
**Urine protein, mg/L**	85.4 ± 199.1	14.9 ± 22.6	56.2 ± 144.6	206.2 ± 307.7	<0.001
**Normalised to E_cr_ as E_x_/E_cr_**					
**E_Cd_/E_cr_, µg/g creatinine**	10.43 ± 8.02	10.26 ± 10.35	9.98 ± 6.79	11.69 ± 9.20	0.641
**E_β2M_/E_cr_, mg/g creatinine**	4.87 ± 16.55	0.23 ± 0.37	1.66 ± 9.72	16.13 ± 27.49	<0.001
**E_α1M_/E_cr_, mg/g creatinine**	11.34 ± 15.00	5.78 ± 4.95	7.53 ± 6.30	24.72 ± 24.57	<0.001
**E_Alb_/E_cr_, mg/g creatinine**	23.21 ± 55.07	10.47 ± 15.68	20.71 ± 59.50	37.88 ± 57.23	<0.001
**E_Prot_/E_cr_, mg/g creatinine**	78.25 ± 174.96	16.73 ± 24.54	57.98 ± 149.26	170.13 ± 246.01	<0.001
**Normalized to C_cr_ as E_x_/C_cr_**					
**(E_Cd_/C_cr_) × 100, µg/L filtrate**	11.27 ± 9.89	8.10 ± 9.06	9.67 ± 6.60	17.44 ± 14.17	<0.001
**(E_β2M_/C_cr_) × 100, mg/L filtrate**	7.74 ± 29.06	0.18 ± 0.28	1.82 ± 11.58	27.82 ± 52.20	<0.001
**(E_α1M_/C_cr_) × 100, mg/L filtrate**	15.00 ± 28.25	4.46 ± 3.59	7.45 ± 6.63	41.20 ± 48.68	<0.001
**(E_Alb_/C_cr_) × 100, mg/L filtrate**	29.06 ± 75.93	7.50 ± 9.83	20.23 ± 56.82	65.68 ± 119.75	<0.001
**(E_Prot_/C_cr_) × 100, mg/L filtrate**	109.9 ± 316.8	13.0 ± 19.1	56.3 ± 141.0	310.2 ± 568.2	<0.001

E_x_, excretion of x; cr, creatinine; C_cr_, creatinine clearance; Cd, cadmium; β_2_M, β_2_-microglobulin; α1M, α1-microglobulin; alb, albumin; Prot, protein. Numbers are the arithmetic means ± standard deviation (SD) [47]. eGFR was based on the CKD–Epidemiology Collaboration equations [48].

**Table 3 biomedicines-12-00718-t003:** Dose–response assessment using E_cr_-normalised data.

Independent Variables	Albuminuria	β_2_-microglobulinuria	Low eGFR
	POR (95% CI)	POR (95% CI)	POR (95% CI)
**Age, years**	1.053 (1.024, 1.082) ***	1.008 (0.988, 1.030)	1.143 (1.104, 1.184) ***
**BMI, kg/m^2^**	1.009 (0.935, 1.089)	0.987 (0.937, 1.039)	1.073 (0.982, 1.173)
**Gender**	0.959 (0.534, 1.722)	0.974 (0.640, 1.484)	0.904 (0.438, 1.846)
**Smoking**	1.903 (1.021, 3.547)	1.087 (0.720, 1.1641)	1.232 (0.583, 2.605)
**Hypertension**	1.815 (1.051, 3.134)	1.262 (0.867, 1.839)	1.474 (0.750, 2.894)
**E_Cd_/E_cr_, µg/g creatinine**			
**<2**	Referent	Referent	Referent
**2−4.99**	0.799 (0.402, 1.586)	1.120 (0.687, 1.826)	1.959 (0.928, 4.133)
**5−9.99**	1.080 (0.526, 2.216)	1.417 (0.875, 2.296)	3.463 (1.466, 8.179) **
**≥10**	1.093 (0.380, 3.139)	1.807 (0.910, 3.587)	3.382 (0.862, 13.27)

POR, prevalence odds ratio; CI, confidence interval. ** *p* = 0.005; *** *p* < 0.001 [49].

**Table 4 biomedicines-12-00718-t004:** Dose–response assessment using C_cr_-normalised data.

Independent Variables/Factors	Albuminuria	β_2_-microglobulinuria	Low eGFR
	POR (95% CI)	POR (95% CI)	POR (95% CI)
**Age, years**	1.050 (1.021, 1.079) **	1.008 (0.986, 1.030)	1.135 (1.094, 1.178) ***
**BMI, kg/m^2^**	1.017 (0.946, 1.093)	0.989 (0.939, 1.043)	1.083 (0.984, 1.192)
**Gender**	1.196 (0.676, 2.116)	0.962 (0.627, 1.474)	1.258 (0.576, 2.744)
**Smoking**	2.009 (1.118, 3.619) *	1.011 (0.667, 1.534)	1.280 (0.589, 2.782)
**Hypertension**	1.912 (1.129, 3.237) *	1.328 (0.907, 1.945)	2.063 (0.992, 4.294)
**(E_Cd_/C_cr_) × 100,** **µg/L filtrate**			
**<2**	Referent	Referent	Referent
**2−4.99**	1.764 (0.886, 3.514)	1.914 (1.100, 3.330) *	5.704 (2.414, 13.48) ***
**5−9.99**	1.950 (1.009, 3.766) *	1.744 (1.030, 2.951) *	10.35 (4.160, 25.76) ***
**≥10**	2.849 (1.136, 7.146) *	2.462 (1.320, 4.595) **	18.06 (3.702, 88.15) ***

POR, prevalence odds ratio; CI, confidence interval. * *p* = 0.016–0.047; ** *p* = 0.001–0.005; *** *p* < 0.001 [49].

**Table 5 biomedicines-12-00718-t005:** Representative rates of filtration, excretion, catabolism and transcytosis in normal and Cd-intoxicated proximal tubular cells.

PTC Status	Protein	Filtration Rate	Excretion Rate	Catabolic Rate	Transcytosis Rate
**Normal**	Albumin	60 gm/d	20 mg/d	2.980 gm/d	57 gm/d
	β2M	300 mg/d	100 μg/d	299.9 mg/d	0
**Cd intoxicated**	Albumin	60 gm/d	50 mg/d	2.950 gm/d	57 gm/d
	β2M	300 mg/d	1000 μg/d	299 mg/d	0

Assumptions: plasma albumin is 40 gm/L; plasma β2M is 2.0 mg/L; GFR is 150 L/d; the glomerular sieving coefficient for albumin (GSCalb) is 0.01; and GSCβ2M is 1.

**Table 6 biomedicines-12-00718-t006:** Cadmium accumulation in human organs.

Country of Origin	Cadmium Content, µg/g Wet Tissue Weight	Reference
Australia, Autopsy, n 61, 2–89 years	The percentage of kidney Cd content ≥ 50 µg/g was 3.3%. ^a^ Mean lung, liver and kidney Cd were 0.13, 0.95 and 15.45 µg/g, respectively. Mean kidney Cd was 16 times higher than that in the liver. Peak hepatic and renal Cd levels were 1.5 and 25.9 µg/g.	Satarug et al. [15]
United Kingdom, Autopsy, n 2700, nationwide (1978–1993)	The percentage of kidney Cd content ≥ 50 µg/g was 3.9%. Mean kidney Cd content was 19 µg/g. Peak renal Cd level was 23 µg/g.	Lyon et al. [95]
Canada (Quebec) Autopsy, n 314	Respective mean liver (kidney) Cd in smokers, ex-smokers and non-smokers were 2.5 (34.5), 1.4 (20.3) and 0.7 (7.0) µg/g. Mean liver Cd in female smokers was higher than male smokers (3.6 vs. 2.2 µg/g). Peak hepatic and renal Cd levels were 2.2 and 44.2 µg/g.	Benedetti et al. [96]
Greenland Autopsy, n 95, 19–89 years	Mean (range) liver Cd content was 5.3 (0.3–24.3) μg/g. Mean (range) kidney Cd content was 43.8 (6.7–126) μg/g. Peak hepatic and renal Cd levels were 1.97 and 22.3 μg/g.	Johansen et al. [97]
Sweden Kidney transplant donors, n 109, 24–70 years, median age 51.	Median kidney Cd was 12.9 μg/g. In non-smokers, the renal Cd accumulation rate was 3.9 μg/g for every 10-year increase in age. An additional 3.7 μg/g accumulation rate for every 10-year increase in smoking. In women who had serum ferritin levels ≤ 20 µg/L (depleted iron stores), the renal Cd accumulation rate was 4.5 μg/g for every 10-year increase in age.	Barregard et al. [98]

^a^ Kidney Cd content of 50 µg/g was used as a toxicological reference value [99].

## Data Availability

Not applicable.

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
