# Peer review of "Is Chronic Kidney Disease Due to Cadmium Exposure Inevitable and Can It Be Reversed?"

_biomedicines, 2024, doi:10.3390/biomedicines12040718_

Round 1

Reviewer 1 Report

Comments and Suggestions for Authors

The review titled: “Is Chronic Kidney Disease due to Cadmium Exposure Inevitable and Can It Be Reversed? ” reports several data about CKD and it arouses considerable interest since highlight mechanisms induced by cadmium. The interest in heavy metals represents an important topic given the diffusion of heavy metals in the environment and its bioaccumulation in higher organisms.

Although the review is very complete, the authors should discuss the topic of heavy metal bioaccumulation at cellular and subcellular levels, citing the relevant literature.

Author Response

Reviewer 1.

Comments and Suggestions

The review titled: “Is Chronic Kidney Disease due to Cadmium Exposure Inevitable and Can It Be Reversed? ” reports several data about CKD and it arouses considerable interest since highlight mechanisms induced by cadmium. The interest in heavy metals represents an important topic given the diffusion of heavy metals in the environment and its bioaccumulation in higher organisms.

Although the review is very complete, the authors should discuss the topic of heavy metal bioaccumulation at cellular and subcellular levels, citing the relevant literature.

RESPONSE: Thank you for evaluating my work, comments, and the suggestion to include the role of the endoplasmic reticulum.   

  • A subtitle, Disruption of Calcium Homeostasis has been added to reflect the trafficking of Cd to the endoplasmic reticulum where it exerts toxicity (lines 526-534), quoted below.

“Disruption of calcium homeostasis has been suggested to be another mechanism by which Cd induced tubular cell death [131-133]. In primary rat proximal tubular cells, Cd caused calcium release from the endoplasmic reticulum, which raised intracellular calcium concentrations, and inhibited autophagy [131,132]. In mouse renal tubular cells, Cd appeared to activate at least two calcium channels, namely phospholipase C (PLC)-inositol 1, 4, 5-trisphosphate receptor (IP3R) and the sarco/endoplasmic reticulum Ca2+-ATPase (SERCA), known to be involved in the release and reuptake of calcium by the ER [132]. Cd suppressed also the SERCA expression and diminished the stability of SERCA protein in mouse tubular epithelial cells, thereby intensifying toxic effects of Cd [133].”

[131] Ning, B.; Guo, C.; Kong, A.; Li, K.; Xie, Y.; Shi, H.; Gu, J. Calcium signaling mediates cell death and crosstalk with autophagy in kidney disease. Cells 2021, 10, 3204.

[132].Liu, F.; Li, Z.F.; Wang, Z.Y.; Wang, L. Role of subcellular calcium redistribution in regulating apoptosis and autophagy in cadmium-exposed primary rat proximal tubular cells. J. Inorg. Biochem. 2016, 164, 99-109.

[133] Li, K.; Guo, C.; Ruan, J.; Ning, B.; Wong, C.K.-C.; Shi, H.; Gu, J. Cadmium disrupted ER Ca2+ homeostasis by inhibiting SERCA2 expression and activity to induce apoptosis in renal proximal tubular cells. Int. J. Mol. Sci. 2023, 24, 5979.

Reviewer 2 Report

Comments and Suggestions for Authors

In the manuscript: “Is Chronic Kidney Disease due to Cadmium Exposure 2 Inevitable and Can It Be Reversed?”, the author reviewed CKD in populations environmentally exposed to Cd. An interesting topic that contributes to knowledge in the area, but certain issues must be corrected.

Major revisions

1.    In the abstract, the objective of the manuscript must be mentioned. Also, authors must mention the gap that the manuscript will fill within the current knowledge in the abstract and the introduction.

2.    Each section must be improved by adding an explanation at the beginning of their description.

3.     Each section must have concluded. That is, the author must resume and conclude each section: for instance, the data above suggest… or the information concluded that…

Minor revisions

1.             Define all the abbreviations presented in the text and do so in their first appearance, either in the abstract or in the body of the manuscript. For example, NOAEL.

2.             Revise grammar.

Comments on the Quality of English Language

no comments

Author Response

Reviewer 2:

Comments and Suggestions

In the manuscript: “Is Chronic Kidney Disease due to Cadmium Exposure Inevitable and Can It Be Reversed?”, the author reviewed CKD in populations environmentally exposed to Cd. An interesting topic that contributes to knowledge in the area, but certain issues must be corrected.

RESPONSE: Thank you for reviewing my work, comments, and suggestions for improvement. All suggestions have been followed through. Change to the text are in blue.

Major revisions

Comment 1.  In the abstract, the objective of the manuscript must be mentioned. Also, authors must mention the gap that the manuscript will fill within the current knowledge in the abstract and the introduction.

RESPONSE:

  • Abstract and introduction have been rewritten to include the objective and the knowledge gap.

Comment 2. Each section must be improved by adding an explanation at the beginning of their description.

RESPONSE:

  • A brief introduction has been added to each section and some subsections.

Comment 3.     Each section must have concluded. That is, the author must resume and conclude each section: for instance, the data above suggest… or the information concluded that…

RESPONSE:

  • This review has undergone some structural changes and it now has 5 sections, and a summary or conclusion has been provided for each section.

Minor revisions

  1. Define all the abbreviations presented in the text and do so in their first appearance, either in the abstract or in the body of the manuscript. For example, NOAEL.

Response:  All abbreviations have been defined in their first appearance.

  1. Revise grammar.

Response:  The grammar corrections/revisions have been undertaken.

Reviewer 3 Report

Comments and Suggestions for Authors

Comments about abstract:  The abbreviation should be first written in the full form.

I fail to understand the purpose or aim of this review. I suggest focusing on the purpose of the review and avoiding excessive general information.

Please begin with a brief introduction of Chronic Kidney Disease, followed by the prevalence of Chronic Kidney Disease in the population and the role of Cadmium. Finally, address or provide updates on the issues you want to discuss in this review.

Comments about introduction: The introduction section should include the signs and symptoms of CKD and should include more literature about CKD and Cd. The introduction should be enriched. The goal of the review should be clear and concise.

Comments on the Quality of English Language

There are many issues with English language of the manuscript.

Author Response

Reviewer 3.

Comments and Suggestions

Comments about abstract:  The abbreviation should be first written in the full form.

I fail to understand the purpose or aim of this review. I suggest focusing on the purpose of the review and avoiding excessive general information.

Please begin with a brief introduction of Chronic Kidney Disease, followed by the prevalence of Chronic Kidney Disease in the population and the role of Cadmium. Finally, address or provide updates on the issues you want to discuss in this review.

RESPONSE:  Thank you for reviewing my work, comments, and suggestions for improvement. All suggestions have been followed through and changes to the text are in blue.

  • All abbreviations have been defined in their first appearance.
  • Abstract and introduction have been rewritten to include the objectives and the knowledge gaps.
  • This review has undergone extensive revisions and structural changes, and now it contains 5 Sections.
  • A brief introduction has been provided for each section.
  • A summary or conclusion has been provided for each section.

Comments about introduction: The introduction section should include the signs and symptoms of CKD and should include more literature about CKD and Cd. The introduction should be enriched. The goal of the review should be clear and concise.

  • The introduction has been rewritten to begin with a brief epidemiology of CKD, followed by Cd and its association with risk of CKD.
  • The objectives of a review have been explicitly stated (lines 58-67), as quoted below.

“This review has three major aims. Firstly, to provide an update on the exposure levels to Cd in the general population that were found to be associated with an enhanced risk of CKD. Special emphasis will be given to the mechanisms by which Cd reduces the absorption of albumin, leading to albuminuria. This has not been clearly defined. Secondly, to demonstrate the impact of a conventional method of normalization of the excretion rates of Cd and albumin to creatinine excretion (Ecr) on dose-response and health risk analyses of Cd. Thirdly, to illustrate that current exposure guidelines do not provide enough of a margin to protect health. To this end, the eGFR reduction and albuminuria are proposed to be adverse health outcomes of chronic Cd exposure from which protective exposure guidelines should be formulated.”

Comments on the Quality of English Language

There are many issues with English language of the manuscript.

RESPONSE:

  • I have sought advice, comments, and edits of the English language from a native English speaker colleague. Typo and grammar errors have been corrected. I hope, the English language of this review has been improved.

Round 2

Reviewer 3 Report

Comments and Suggestions for Authors

The authors have made suggested improvements.